# The effect of lidocaine against sepsis-induced acute lung injury in a mouse model through the JAK2/STAT3 pathway

Ying Cai, Hong Zhang, Lun-Meng Cui, Qian Chen, Feng-Jie Xie [ID]*

Department of Critical Care Medicine, Hongqi Hospital affiliated to Mudanjiang Medical University, Mudanjiang City, Heilongjiang Province, China

* mdjxfj1971@sina.cn

## Abstract

### Objective

This study aimed to investigate the effects of lidocaine on sepsis-induced acute lung injury and its underlying mechanisms.

### Methods

Thirty C57BL/6 mice were divided into three groups: SHAM, CLP, and LD. The sepsis-induced acute lung injury model was established using cecal ligation and puncture (CLP) surgery, while SHAM mice underwent a sham operation without ligation or puncture. Mice in the LD group were administered lidocaine (10 mg/kg) intravenously through the tail vein. The SHAM and CLP groups were treated with an equal volume of 0.9% sterile saline solution. All mice were sacrificed 24 hours after surgery, and lung tissue and blood samples were collected for subsequent analysis. The wet/dry weight ratio (W/D ratio) was measured to evaluate lung edema. Lung injury and apoptosis were assessed using HE staining and TUNEL assay. The concentrations of inflammatory cytokines IL-6, TNF-α, and HMGB1 were measured by ELISA. The expression of JAK2, STAT3, p-STAT3, Bcl-2, HMGB1, and Bax was analyzed by western blot.

### Results

The W/D ratio in the CLP group was significantly higher than the SHAM group, indicating increased lung edema. Pathological examination revealed obvious lung injury, and apoptosis was evident in the CLP group. The expression of HMGB1, IL-6, and TNF-α in lung tissue increased by 24 hours after CLP surgery. Additionally, the levels of JAK2, STAT3, p-STAT3, HMGB1, and Bax were significantly increased, while Bcl-2 expression was reduced. However, lidocaine administration reversed these changes.

**Data availability statement:** All relevant data are within the manuscript and its Supporting Information files.

**Funding:** This study was financially supported by the National Clinical Key Specialty Construction Project in the form of a special fund of the central financial subsidy received by FX. No additional external funding was received for this study.

**Competing interests:** The authors have declared that no competing interests exist.

## Conclusion

Intravenous lidocaine effectively alleviated acute lung injury in septic mice. The anti-inflammatory effects of lidocaine may be attributed to its suppression of the JAK2/STAT3 signaling pathway and its anti-apoptotic effects.

## Introduction

The most common cause of death for patients in intensive care units is sepsis, which can be defined as a dysregulated host response to infection that can result in irreversible organ dysfunction and death [1]. The incidence of sepsis-induced acute lung injury as well as the associated mortality are high. At present, respiratory technology is commonly used in clinical support treatment. However, there is no specific drug, so it is imperative to find relevant therapeutic drugs and explore their underlying mechanisms.

Janus kinase signal transducer and activator of transcription (JAK/STAT) pathway is a critical cellular signal transduction pathway and an important pathway for cytokine signal transduction in the pathogenesis of sepsis, such as tumor necrosis factor-α (TNF-α) and interleukin-6 (IL-6), among others. Studies have shown that JAK2 can be rapidly activated when sepsis occurs. The nuclear translocation of STAT3 in lung tissue is significantly increased, and it can be detected 2 h after cecal ligation and puncture (CLP), indicating that the JAK/STAT signaling pathway plays a regulatory role in multiple-organ damage in CLP-induced septic rats [2].

Lidocaine is one of the most commonly used local anesthetics in clinical practice. Studies have demonstrated that lidocaine can effectively reduce the concentration of high-mobility group box 1 (HMGB1) in the serum of patients after radical hysterectomy [3]. Furthermore, intravenous infusion of lidocaine during the perioperative period was found to reduce the inflammatory factors related to lung injury in the serum of patients with esophageal cancer after surgery [4]. As HMGB1 is a key mediator of late inflammatory cytokines and severe sepsis, it could be speculated that the protective effect of lidocaine on lung injury may be to decrease the inflammatory level of septic patients by inhibiting the release of HMGB1. Several studies have confirmed that lidocaine can exert its effect against acute lung injury by regulating different anti-inflammatory pathways [5–7]. However, whether it is related to inhibiting the activation of the JAK/STAT signaling pathway remains to be ascertained. The present study, based on the mouse models of sepsis-induced acute lung injury, explored whether lidocaine exerts an anti-inflammatory effect by regulating the JAK2/STAT3 pathway and whether it can affect the apoptosis of lung cells.

## 1. Materials and methods

### 1.1 Materials

During the material collection stage, 2% lidocaine hydrochloride was obtained from Shandong Huachen Pharmaceutical Co., Ltd. HMGB1 Mouse ELISA Kit, IL-6 Mouse ELISA Kit, and TNF-α Mouse ELISA Kit were obtained from Quanzhou Ruixin

Biotech Co., Ltd. Anti-mouse JAK2, STAT3, p-STAT3, Bcl-2, and HMGB1 were obtained from Abcam (the United States). The animal study was reviewed and approved by the Animal Ethics Committee of Mudanjiang Medical University.

## 1.2 Experimental grouping and establishment of the septic model in mice

Thirty healthy C57bl/6 male mice of SPF grade 8–12 weeks old, with a body mass of 18–22 g, were provided by the Animal Experiment Center of Mudanjiang Medical College (SYXK 2019–003 Heilongjiang, China). They were divided into three groups According to the random number scale (n = 10): sham operation group (SHAM group), model group (CLP group), and treatment group (LD group). This experiment was approved by the Experimental Animal Ethics Committee of Mudanjiang Medical College, and all operation processes strictly followed the welfare ethics of animal experiments.

The animals were fed with a standard diet and water ad libitum, housed at constant temperature (20–25°C), with alternating 12 h of light and darkness. Ten minutes before the laparotomy and 0, 1, and 2 h after the operation, the mice in the SHAM group and CLP group were injected with 0.9% normal saline (10 mg/kg) and lidocaine (10 mg/kg) into the tail vein, respectively. The surgical procedure is summarized as follows. First, the mice were anesthetized with isoflurane. Second, the middle of the abdomen was cut, and the cecum was ligated at a distance of 1/2 from the root of the cecum after exposing it. Next, the cecum was punctured with a 20 # needle. After that, the cecum was returned to the abdominal cavity, and the abdominal cavity was sutured layer by layer. After the operation, sterile saline preheated to 37°C was immediately injected subcutaneously. The same operation was performed on SHAM mice, but cecal ligation and perforation were not performed. All mice were injected subcutaneously with sterile saline preheated to 37°C 6 h after operation for volume resuscitation.

After the model was established, the mice developed listlessness, huddling behavior, pilar erecti, and corner discharge of the eye, which exacerbated over time. In addition, the model was successfully established according to the increase in inflammatory factors in mouse serum and lung,the W/D ratio of the lung tissues, and the typical inflammatory lesions of pathological sections.

## 1.3 Specimen collection

After 24 h, all mice were euthanized under anesthesia induced by isoflurane. The thorax was opened, and blood was collected from the left ventricle, immediately placed on ice, and centrifuged, and the upper serum was taken and frozen in the refrigerator at −20°C. Lung tissue was taken from the left lower lobe of the rat lung and fixed with 4% paraformaldehyde. The remaining lung lobe samples continued to the next step after cleaning the residual blood. The flowchart outliningthe entire experimental procedure is depicted in Fig 1.

## 1.4 Indicator detection

**1.4.1 The measurement of the lung wet/dry weight (W/D) ratio assay.** After absorbing the exudate and blood stain on the surface of the upper lobe of the left lung tissue with filter paper, the wet weight was measured and then placed in an 80°C incubator for continuous drying for 72 h to attain a constant weight. The dry weight was measured again. The wet weight/ dry weight ratio (W/D) = wet weight/ dry weight × 100% and the water content of lung tissue was determined.

**1.4.2 Histopathological examination of the lungs and TUNEL assay to detect cell apoptosis.** After formalin fixation for 24 h, dehydration, and paraffin embedding, tissue slices were prepared and H&E-stained. The changes in cellular morphology were assessed using light microscopy. Paraffin sections were also used to detect apoptosis using the TUNEL method.

**1.4.3 Serum cytokine measurement to assess the serum levels of HMGB1 and the lung tissue of HMGB1, IL-6, and TNF-α.** The ELISA analysis was carried out to determine HMGB1 concentration in the frozen serum. The lung tissue of mice in each group was accurately weighed, and the protease inhibitors were added. After homogenization at a low

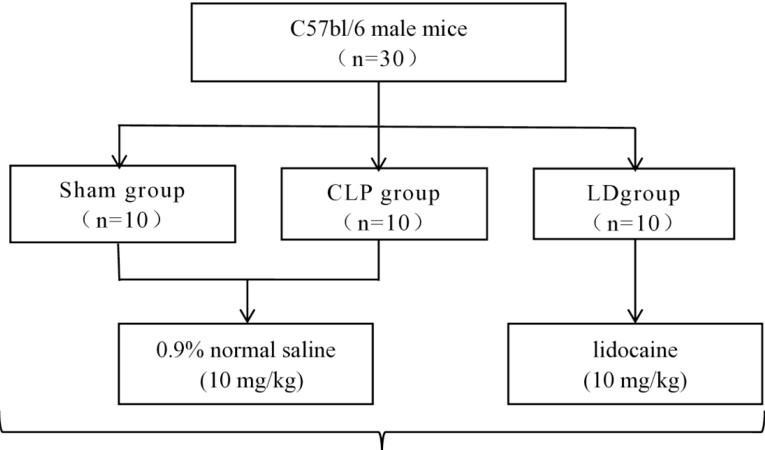

**Fig 1. The diagram of experimental design and grouping.** Note: terminal deoxynucleotidyl transferase dUTP Nick End Labeling (TUNEL), High-mobility group box 1 protein(HMGB1), interleukin-6(IL-6), tumor necrosis factor-α (TNF-α), Janus kinase 2(JAK2), Signal transducer and activator of transcription 3(STAT3), phosphorylated Signal transducer and activator of transcription 3(p-STAT3), B-cell lymphoma 2(Bcl-2), Bcl-2-associated X protein(Bax) .

temperature, the supernatant was used for detecting the concentration of HMGB1, IL-6, and TNF-α in accordance with the instructions specified in the Mouse Insulin ELISA kit.

**1.4.4 Western blot analysis of JAK2, STAT3, p-STAT3, Bcl-2, HMGB1, and Bax.** The lung tissues of mice in each group were weighed and then lysed with lysate at 4°C after cryogenic grinding. The protein concentration was determined with a BCA kit. The total proteins in denatured samples were separated by gel electrophoresis, then blocked, and then incubated with the primary antibody overnight at 4°C, followed by three washes with 1X TBST. Samples were then incubated with the secondary antibody for 70 min and washed with 1X TBST again. Finally, membranes were exposed using ECL reagents, and the gray level was calculated after image acquisition.

## 1.5 Statistical analysis

Experimental data analyses were performed with IBM SPSS Statistics 26.0 software. Normality test and homogeneity of variance analysis were performed first. One-way analysis of variance were used for those that satisfied the predetermined conditions, and non-parametric test were used for those that did not satisfy the conditions. A p-value of $< 0.05$ indicated that the difference was statistically significant.

## 2. Results

### 2.1 Lidocaine alleviated acute lung injury in sepsis

Twenty-four hours after CLP in mice, the lung tissue showed serious edema, while the treatment of lidocaine reduced the pulmonary edema (Fig 2, $p < 0.01$). Histopathological observations are described as follows. In the SHAM group,

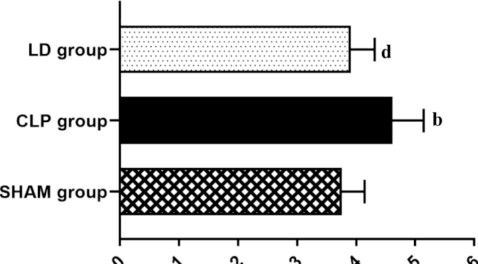

**Fig 2. Wet/dry weight ratio of lung tissue in each group.** Note: [a]$P<0.05$, [b]$P<0.01$ vs SHAM group, [c]$P<0.05$, [d]$P<0.01$vs. CLP group.

the tracheal wall was intact and normal, the tissue was neatly arranged with no hyperemia or edema, and there was no obvious inflammatory cell infiltration. In the CLP group, the tracheal wall was incomplete, the alveolar septa were wide with capillary congestion, edema, and visible inflammatory cell infiltration. In the CL group, the tracheal wall suffered from varying degrees of structural damage, but apparently, the changes were less pronounced than those in the CLP group (Fig 3). Subsequently, through the modified semi-quantitative histopathological score system in all groups, it was found that the pathological scores of the CLP group and LD group were significantly higher than that of the SHAM group, while the pathological score of the LD group was definitely lower than that of the CLP group (Fig 2, $p<0.01$).

## 2.2 Lidocaine alleviated sepsis-induced acute lung injury by reducing apoptosis

TUNEL staining results revealed that compared with the SHAM group, the percentage of apoptotic cells in the CLP group increased significantly (Fig 4, $p<0.05$). Lidocaine treatment exerted a therapeutic effect, suggesting that lidocaine could reduce the inflammatory reaction induced by CLP and had a significant preventive and therapeutic effect in the

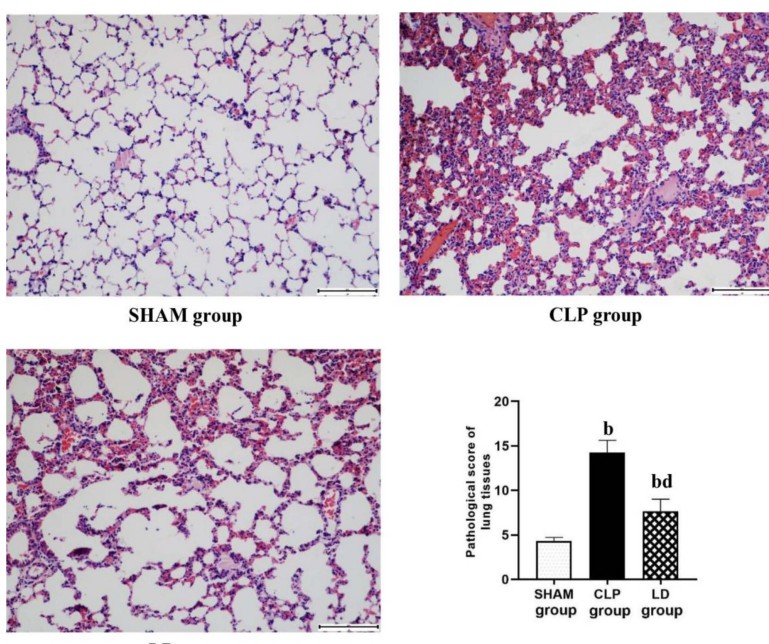

**Fig 3. HE staining sections and pathological scores of lung tissue(200 ×).** Note: [a]$P<0.05$, [b]$P<0.01$ vs SHAM group, [c]$P<0.05$, [d]$P<0.01$vs. CLP group.

CLP-induced acute lung injury model. We speculate that the relevant mechanism may involve the apoptosis-associated signaling pathways by resveratrol.

## 2.3 Lidocaine down-regulated the pro-inflammatory in serum and lung tissue of septic mice

The ELISA results showed that compared with the SHAM group, the level of HMGB1 in serum and lung tissue of the CLP group mice was significantly increased, while lidocaine treatment could down-regulate the level of the above inflammatory factors (Fig 5. A, p<0.05). Simultaneously, the expression levels of TNF-α and IL-6 increased greatly in lung tissue of CLP group mice, implying that lidocaine treatment could effectively inhibit TNF-α and IL-6 induced by CLP (p<0.01).

## 2.4 Lidocaine inhibited the activation of the JAK2/STAT3 signaling pathway in lung tissue of septic mice

To further address the anti-inflammatory effect of lidocaine, testing for several key proteins in the JAK2/STAT3 signaling pathway was carried out. We observed that compared with the SHAM group, the expression levels of JAK2, STAT3,

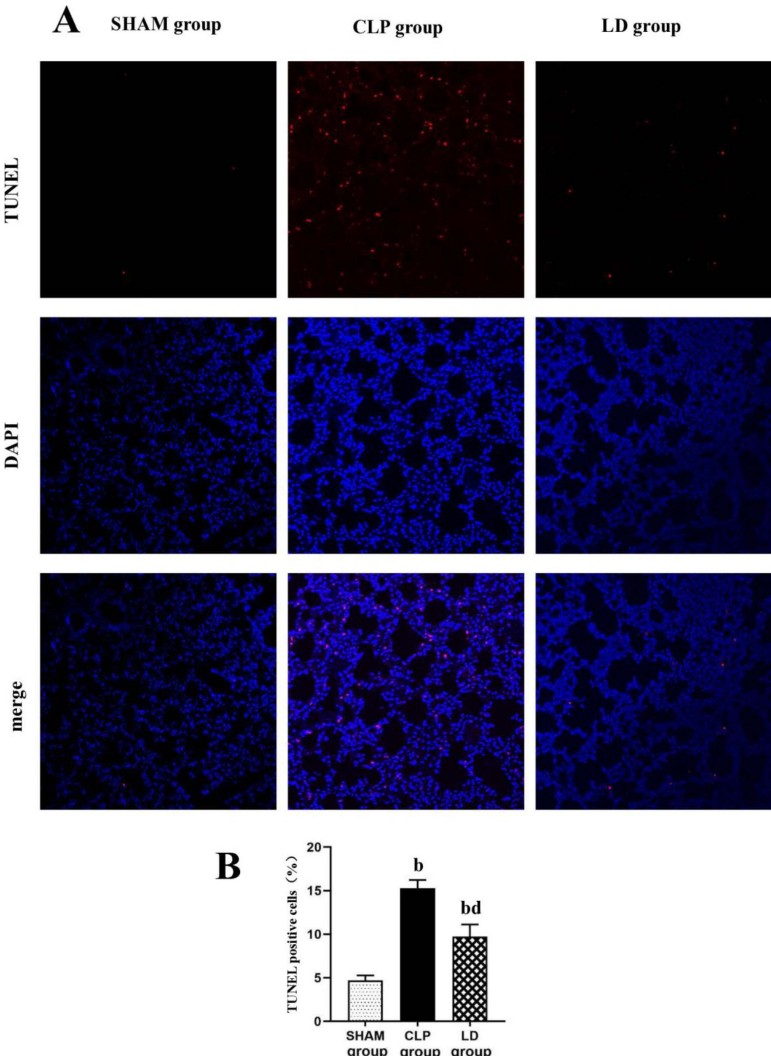

**Fig 4. TUNEL fluorescence staining of lung tissue.** Note: A. Representative images for TUNEL staining in lung tissue (red: TUNEL positive cells, blue: DAPI, ×200). B.Statistical analysis results of TUNEL staining. [a]P<0.05, [b]P<0.01 vs SHAM group, [c]P<0.05, [d]P<0.01vs. CLP group.

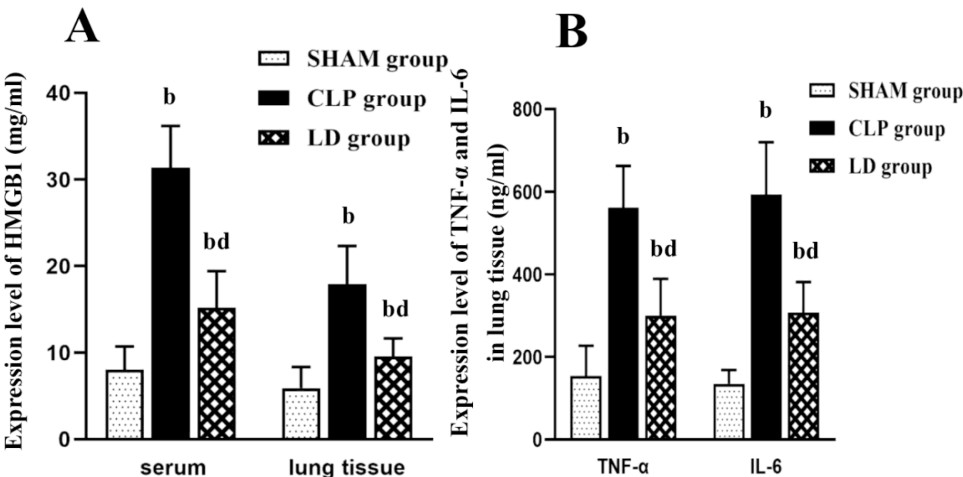

**Fig 5. Relative expression levels of HMGB1, IL-6 and TNF-α detected via ELISA.** Note: [a]$P<0.05$, [b]$P<0.01$ vs SHAM group, [c]$P<0.05$, [d]$P<0.01$vs. CLP group.

p-STAT3, HMGB1, and Bax in the lung tissue of CLP group mice were significantly increased, while the level of Bcl-2 was decreased (Fig 6, $p<0.05$). However, the changes in the above protein levels could be reversed by lidocaine treatment. Moreover, after calculating the ratio of Bax/Bcl-2, we realized that lidocaine treatment also reduced the Bax/Bcl-2 ratio (Fig 6, $p<0.05$), proving that lidocaine could exert anti-inflammatory effect by inhibiting the activation of JAK2/STAT3 signaling pathway possibly.

## 3. Discussion

Cytokines belong to functioning proteins with small molecules. They have a very short half-life of a few minutes to a few hours but play a critical role in the inflammatory process. During sepsis, the massive release of pro-inflammatory cytokines can activate a variety of cell signal transduction pathways and mediate the post-inflammatory cascade reaction, leading to the amplification of inflammatory response and even triggering an inflammatory storm, causing uncontrolled inflammation and multiple-organ failure. Cytokines can mediate inflammation by using complex signal cascades, thus exerting their biological effects.

In this study, the results showed that the serum HMGB1 of CLP model group mice increased sharply 24 h after lidocaine treatment, and the lung tissue HMGB1, IL-6, and TNF-α also increased dramatically. Lidocaine treatment inhibited the increase in the above pro-inflammatory cytokines. HMGB1 is a nuclear protein widely recognized as a transcription factor and growth factor, which was later identified as a key mediator of severe sepsis [8,9]. In addition, as one of the late inflammatory factors, HMGB1 can also be released from inflammatory cells and necrotic tissues [10], so the excessive release of HMGB1 plays a key role in the pathogenesis of acute and chronic inflammation. It has been confirmed that HMGB1 binds to cell surface receptors once it is released from the nucleus. In addition, serum HMGB1 levels begin to rise significantly 18 h after surgery to induce peritonitis, while specific inhibition of HMGB1 activity 24 h after surgery can significantly improve the survival rate of mice [11]. It can be seen that in terms of anti-inflammatory treatment of sepsis, the therapeutic window of anti-HMGB1 may be wider compared with the intervention treatment of some early inflammatory factors (such as IL-6 and TNF-α). Our study confirmed that lidocaine could regulate the production and release of HMGB1 to regulate the inflammatory cascade reaction, which may have a guiding significance for the development and application of specific antibodies against HMGB1.

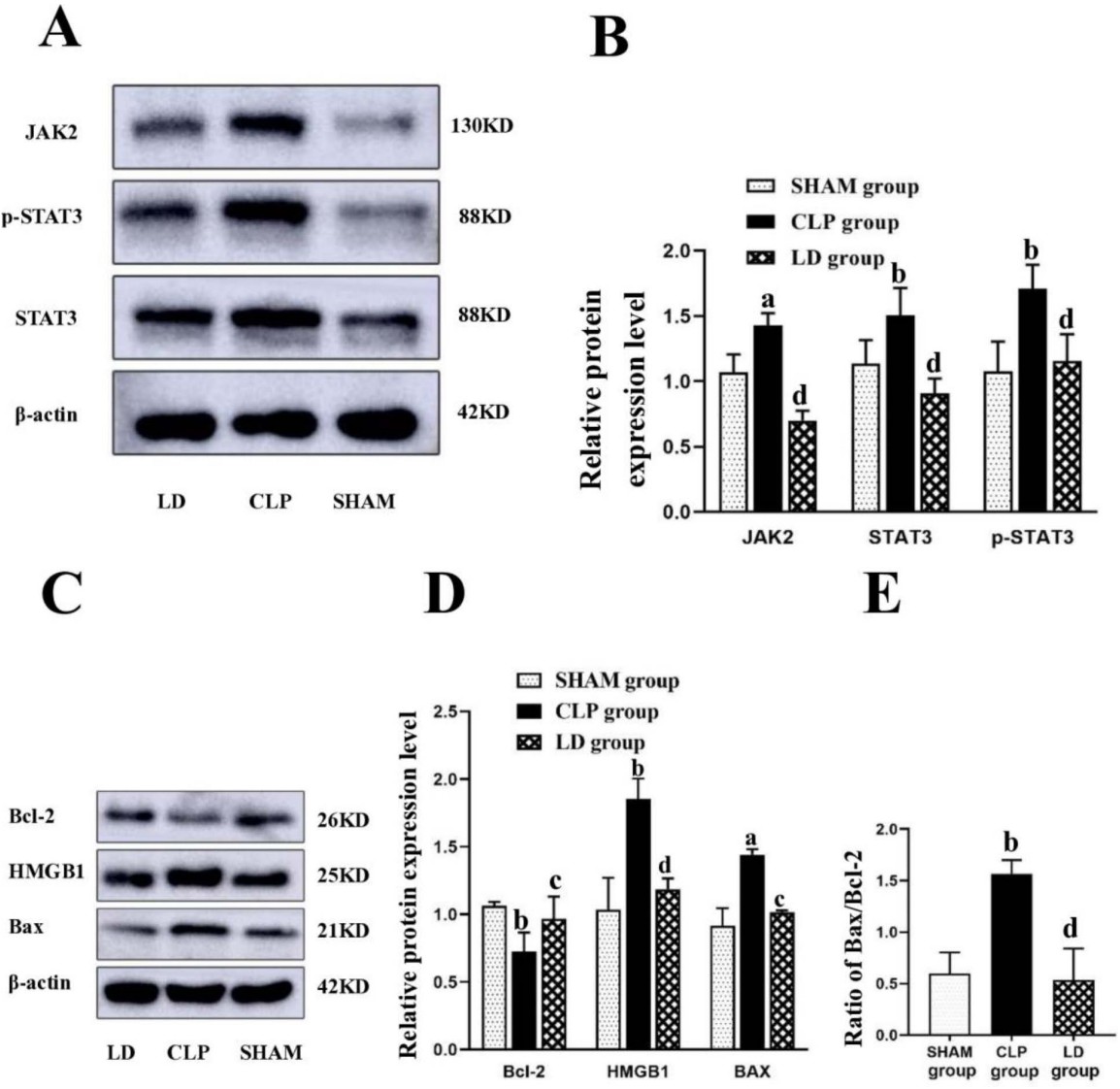

**Fig 6. The expression of relative protein detected with western blot.** Note: [a]P<0.05, [b]P<0.01 vs SHAM group, [c]P<0.05, [d]P<0.01vs. CLP group.

The JAK/STAT pathway is one of the main signaling pathways of sepsis. The key cell signaling factors in this pathway include IL-6 and TNF-α, which we have studied in our experiment [12,13]. Interleukin-6 (IL-6) and Tumor Necrosis Factor-α (TNF-α) are both pivotal cytokines that play crucial roles in immune regulation and inflammatory responses. The results of this study confirm that, compared to the SHAM group, the expression levels of TNF-α and IL-6 in the lung tissue of mice in the cecal ligation and puncture (CLP) group were significantly elevated (P<0.01). Treatment with lidocaine effectively inhibited the CLP-induced increase in TNF-α and IL-6 expression (P<0.01). The general process is to activate the relevant JAK kinase after the combination of inflammatory cytokines and corresponding receptors, thus selectively phosphorylating the STAT protein family, making the activated STAT protein translocate to the nucleus and play a key role in the transcription of target genes. STAT3 is an important member of the STAT family and is closely related to inflammation. When sepsis occurs, the inflammatory factors IL-6 and TNF-α released at the early stage in the body can bind

with STAT3, activate it, and enhance IL-6 and TNF-α transcription in the nucleus. This process can not only transduce inflammatory signals but also regulate and exacerbate inflammation. Many inflammatory factors generated by this process continue to activate the JAK/STAT3 signaling pathway, thus forming a positive feedback loop, further exacerbating inflammation and ultimately leading to uncontrolled inflammation and systemic inflammation [14]. Therefore, effective inhibition of inflammation is the key to the treatment of sepsis.

The results demonstrated that the JAK2/STAT3 signaling pathway was highly expressed in the lung tissue of CLP septic mice. At the same time, during the development of sepsis, the expressions of JAK2 and STAT3 proteins were significantly up-regulated, and the transcription and expression of downstream inflammatory factors IL-6 and TNF-α were also increased. This indicates that the JAK2/STAT3 signaling pathway is involved in the pathogenesis of sepsis and the regulation of inflammation. It has been proven that lidocaine can mitigate the inflammatory cascade reaction and promote tissue repair [15]. However, the effect of lidocaine on the JAK2/STAT3 signaling pathway is somewhat unclear. The results of this study show that intravenous lidocaine intervention is beneficial to improve the lung morphology of septic mice and can inhibit the increase of JAK2 and STAT3 in the lung tissue as well as the expression of inflammatory factors IL-6 and TNF-α.

In this study, we found that the proportion of apoptotic cells in the lung tissue of CLP group mice increased, and intravenous lidocaine could reduce the increase in apoptotic cells. In addition, lidocaine reduced the expression and release of HMGB1 in serum and IL-6 and TNF-α in the lungs of CLP model group mice, suggesting that it has a protective effect on organ damage. In other studies, Andrea et al [16]. observed that intravenous lidocaine reduced the inflammation and apoptosis of lung ischemia–reperfusion. A few studies also suggested that the cardioprotective effect of lidocaine administered before and after ischemic injury could be achieved through anti-apoptosis [17]. These findings suggest that one of the mechanisms of organ-protective effects of lidocaine may be related to the reduction of cell apoptosis and inflammation.

Apoptosis is a form of cell death that is strictly regulated by genes. In the process of sepsis-induced immunosuppression, apoptosis plays a key role in the maintenance of selection and functional immune response of immune cell groups. There are three main apoptosis pathways in mammalian cells: extrinsic death receptor pathway, endogenous mitochondrial pathway, and endoplasmic reticulum stress-induced pathway [18]. Among them, the endogenous mitochondrial pathway is driven by changes in the interactions of Bcl-2 family members, including anti-apoptotic molecules Bcl-2, BclxL, etc. In addition, pro-apoptotic molecules include Bax and Bad, among others. In tissue cells, anti-apoptotic Bcl-2 protein interacts with pro-apoptotic Bax [19].

It Is found that all downstream targets of the STAT3 pathway are involved in cell proliferation and apoptosis, including Bcl-2 and Bax [20], and Bax/Bcl-2 ratio is regarded as a key indicator of apoptosis [21]. In this experiment, it was observed that after 24 h of sepsis-induced lung injury in mice, the expression of the anti-apoptotic protein Bcl-2 decreased in the lung tissue, and the expression of pro-apoptotic protein Bax increased, while the treatment of lidocaine could increase the expression level of Bcl-2 protein in the lung tissue and reduce the expression level of Bax protein and the ratio of Bax/Bcl-2, which also corroborated the findings presented in previous studies that the anti-inflammatory effect of lidocaine might be closely related to anti-apoptosis. Nevertheless, lidocaine plays a controversial role in cell apoptosis. Some reports believe that lidocaine can inhibit apoptosis, while others believe that it can induce apoptosis. However, the latter studies report that lidocaine can effectively promote cell apoptosis, which is generally limited to specific cells such as tumor cells [22]. In terms of inflammatory diseases, our experimental results support that the anti-inflammatory mechanism of lidocaine is anti-apoptosis.

The 2021 international guidelines for the management of sepsis and septic shock recommend the use of anti-inflammatory corticosteroids for specific groups of sepsis patients. Commonly used drugs in clinical practice include dexamethasone, among others [23]. Despite its potent anti-inflammatory effects, dexamethasone can reduce inflammatory exudation, capillary dilation, and cellular phagocytic response. However, as a steroid medication, it can affect metabolism, leading to increased blood glucose, hypertension, and osteoporosis; suppress the immune system, increasing the risk of infection; and cause side effects such as gastric ulcers and gastrointestinal bleeding. High mortality rates are still observed

in ICUs, and there has been a long-standing need for other tools to manage these patients. Therefore, efforts to improve anti-inflammatory treatments are ongoing. Although the anti-inflammatory effects of lidocaine proposed in this study are not as potent as those of dexamethasone, it can avoid the therapeutic risks associated with steroid medications and may offer better immune function and outcomes for critically ill patients. A case report introduced the impact of intravenous lidocaine on a patient with severe pneumonia, observing a significant decrease in D-dimer and CRP levels, indicating a good response to lidocaine treatment without adverse events [24]. However, due to concerns about the potential toxicity of lidocaine, intermittent infusion rather than continuous infusion is often chosen in clinical practice. Another study reported that the combination of lidocaine and dexamethasone can regulate the nuclear factor kappa B (NF-κB) pathway, inflammasome activation, and interferon gamma receptor (IFNγR) signaling [25]. Therefore, these results suggest that even though lidocaine cannot completely replace steroids as the primary anti-inflammatory drug in clinical practice, it can help to reduce the dosage of steroid medications and enhance their therapeutic effects. These alternative administration routes and the reported synergistic effects with corticosteroids may help to improve and expand the options available for managing patients with sepsis complicated by acute lung injury.

To sum up, our results prove that lidocaine can play an anti-inflammatory role in the CLP model of septic mice by inhibiting the inflammatory reaction mediated by JAK2/STAT3 and inhibiting the secretion of inflammatory cytokines. This study may provide a new method for the treatment of inflammation and sepsis. Nevertheless, this study has its limitations, as it employed a single animal model and the short 24-hour observation window post-CLP. Subsequent research should explore the long-term impacts of lidocaine administration and assess its therapeutic efficacy in diverse animal models or under different sepsis conditions. Furthermore, as HMGB1 is an advanced inflammatory factor, the treatment of anti-HMGB1 may be very interesting due to the problem of its treatment window, and some studies have proven that specific HMGB1 antagonists may be effective in the clinical treatment of sepsis [12], and it may be considered to use specific drugs in the future treatment. The imbalance of apoptosis during sepsis and how to regulate its process to protect important organs are also important research ideas at present.

## 4. Conclusion

In this study, we found that lidocaine effectively alleviates acute lung injury in septic mice, and its mechanism may be related to the regulation of the JAK2/STAT3 signaling pathway, downregulation of inflammatory factor expression, and anti-apoptotic effects. This is mainly reflected in the following aspects: (1) Lidocaine may reduce acute lung injury in sepsis by decreasing apoptotic cells; (2) Lidocaine downregulates the expression of signaling molecules such as JAK2, STAT3, p-STAT3, and Bax in the lung tissue of septic mice, thereby inhibiting the activation of the JAK2/STAT3 signaling pathway; (3) Lidocaine reduces the levels of inflammatory factors such as HMGB1, IL-6, and TNF-α in the serum and lung tissue of septic mice, thereby mitigating the inflammatory response. This indicates that lidocaine may have potential therapeutic value for the treatment of lung injury induced by sepsis. Nevertheless, additional studies are required to ascertain the long-term prognostic effects of lidocaine on patients suffering from sepsis.

## Supporting information

**S1 Raw Images.  XXXX.**
(ZIP)

**S2 Raw Images.  XXXX.**
(ZIP)

## Acknowledgments

All authors listed have made substantial, direct, and intellectual contribution to the work and approved it for publication.

## Author contributions

**Data curation:** Hong Zhang, Lun-Meng Cui.

**Writing – original draft:** Ying Cai, Qian Chen.

**Writing – review & editing:** Feng-Jie Xie.

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
