## [Decision Letter · Decision Letter 0]

4 Oct 2024

PONE-D-24-28154The effect of lidocaine against sepsis-induced acute lung injury in a mouse model through the JAK2/STAT3 pathwayPLOS ONE

Dear Dr. Feng-Jie,

Thank you for submitting your manuscript to PLOS ONE. After careful consideration, we feel that it has merit but does not fully meet PLOS ONE’s publication criteria as it currently stands. Therefore, we invite you to submit a revised version of the manuscript that addresses the points raised during the review process.

We look forward to receiving your revised manuscript.

Kind regards,

Yong Jiang, Ph.D.

Academic Editor

PLOS ONE

J

2. To comply with PLOS ONE submissions requirements, in your Methods section, please provide additional information regarding the experiments involving animals and ensure you have included details on methods of sacrifice and efforts to alleviate suffering.

Reviewers' comments:

Reviewer's Responses to Questions

**Comments to the Author**

1. Is the manuscript technically sound, and do the data support the conclusions?

Reviewer #1: Yes

Reviewer #2: Yes

2. Has the statistical analysis been performed appropriately and rigorously? 

Reviewer #1: Yes

Reviewer #2: I Don't Know

3. Have the authors made all data underlying the findings in their manuscript fully available?

Reviewer #1: Yes

Reviewer #2: Yes

4. Is the manuscript presented in an intelligible fashion and written in standard English?

Reviewer #1: Yes

Reviewer #2: Yes

5. Review Comments to the Author

Reviewer #1: The methodology is detailed and well-structured. The description of the animal model and experimental groups is clear. However, including a schematic diagram of the experimental design might help readers visualize the study setup more effectively. Also, more detailed information on the anesthesia and euthanasia protocols for animal welfare considerations should be provided.

The discussion effectively interprets the findings and relates them to the broader context of sepsis and acute lung injury. However, it could be expanded to address potential mechanisms by which lidocaine might influence other signaling pathways related to inflammation and apoptosis. Additionally, discussing the translational potential of these findings to clinical settings would enhance the manuscript’s impact.

While the manuscript is written in generally good English, there are several areas with minor grammatical errors and awkward phrasing. A thorough review and editing for language will improve the manuscript’s readability. Examples include correcting “2% lidocaine hydrochloride was purchased from the Shangdong Huachen Pharmaceutical Co., Ltd.” to “2% lidocaine hydrochloride was obtained from Shandong Huachen Pharmaceutical Co., Ltd.” and revising phrases like “constant temperature (20℃–25℃)” to “constant temperature (20–25℃)”.

Reviewer #2: The study clearly outlines its goal to investigate the effects of lidocaine on sepsis-induced acute lung injury and its mechanisms through the JAK2/STAT3 pathway, which is a valuable contribution to the field of critical care and sepsis research. The authors have performed ELISA, Western blotting, Immunohistochemistry to defend their conclusion. The figures are clear and clearly support the Results. This study has important therapeutic implications with lidocaine use for anti-inflammatory effects. However, I have some concerns mentioned below:

1. The authors must put more attention to explaining the mechanism how this pathway precisely mediates anti-inflammatory effects?

2. The authors must include some other known anti-inflammatory treatments like dexamethasone to compare the results with Lidocaine treatment. That could strengthen the conclusion of this study.

3. The discussion section should be expanded to address the limitations of the study, such as the use of a single animal model and the short 24-hour observation window post-CLP. Future studies could investigate the long-term effects of lidocaine administration and test its efficacy across different animal models or septic conditions.

4. while this study has been done on mouse model how these findings could be translated to clinical practice? What are the short term and long-term side effects of using Lidocaine?

5. The authors should also summarize the Conclusion section more explicitly to address main findings and their broader relevance and they can also add potential next steps in this line of research.

6. PLOS authors have the option to publish the peer review history of their article (what does this mean? ). If published, this will include your full peer review and any attached files.

**Do you want your identity to be public for this peer review?** For information about this choice, including consent withdrawal, please see our Privacy Policy .

Reviewer #1: **Yes: ** Ahmad Reza Dehpour

Reviewer #2: **Yes: ** Nadeem Bhat

---

## [Author Response · Author response to Decision Letter 1]

14 Dec 2024

Firstly, I would like to express my gratitude for your attention and professional dedication to my research work. I appreciate and value your suggestions, and I have taken them seriously, making reasonable modifications. I am now returning the revised manuscript to you with the changes highlighted in red for your further review. Additionally, I will actively answer any questions you may raise, and if there are any inadequacies, please feel free to point them out at any time. Lastly, I would like to thank you once again for your valuable advice and suggestions, and I look forward to further cooperation with you. I am committed to continuously improving the quality of this paper. Below are my responses to and changes based on your comments.

Reviewer #1

1.A schematic diagram of the experimental design has been completed and added to the text.

2.The animals in this experiment were euthanized using inhalant anesthetics, and this has been added to the experimental methods.

3.The potential mechanisms by which lidocaine may affect signaling pathways related to inflammation and apoptosis are further mentioned in the discussion. Additionally, content related to the translational potential of these findings in the clinical setting has also been added.

4.I have carefully corrected the grammatical errors you mentioned.

Reviewer #2

1.This revision provides a more detailed description of how this pathway precisely mediates its anti-inflammatory effects.

2.In the discussion section, content comparing other anti-inflammatory drugs with lidocaine has been added, along with relevant literature citations.

3.The discussion section has been expanded to introduce the limitations of this study and to propose and design the next steps in the research.

4.The connection between this experiment and clinical practice has been further elaborated, and some solutions to the side effects of lidocaine have been introduced through literature. The next experiment will further study the long-term effects of lidocaine on septic mice.

5.The conclusion section has undergone significant revisions, providing a detailed summary of the research conclusions of this experiment, and the next steps in the research have also been mentioned.

---

## [Decision Letter · Decision Letter 1]

9 Mar 2025

PONE-D-24-28154R1The effect of lidocaine against sepsis-induced acute lung injury in a mouse model through the JAK2/STAT3 pathwayPLOS ONE

Dear Dr. Feng-Jie,

Thank you for submitting your manuscript to PLOS ONE. After careful consideration, we feel that it has merit but does not fully meet PLOS ONE’s publication criteria as it currently stands. Therefore, we invite you to submit a revised version of the manuscript that addresses the points raised during the review process.

We look forward to receiving your revised manuscript.

Kind regards,

Yong Jiang, Ph.D.

Academic Editor

PLOS ONE

**Journal Requirements:**

**Additional Editor Comments:**

Minor Revision

Reviewers' comments:

Reviewer's Responses to Questions

**Comments to the Author**

1. If the authors have adequately addressed your comments raised in a previous round of review and you feel that this manuscript is now acceptable for publication, you may indicate that here to bypass the “Comments to the Author” section, enter your conflict of interest statement in the “Confidential to Editor” section, and submit your "Accept" recommendation.

Reviewer #2: All comments have been addressed

2. Is the manuscript technically sound, and do the data support the conclusions?

Reviewer #2: Yes

3. Has the statistical analysis been performed appropriately and rigorously? 

Reviewer #2: I Don't Know

4. Have the authors made all data underlying the findings in their manuscript fully available?

Reviewer #2: Yes

5. Is the manuscript presented in an intelligible fashion and written in standard English?

Reviewer #2: Yes

6. Review Comments to the Author

**Reviewer #2: ** The authors have addressed all the questions raised and therefore, recommend it for publication in the journal.

7. PLOS authors have the option to publish the peer review history of their article (what does this mean? ). If published, this will include your full peer review and any attached files.

**Do you want your identity to be public for this peer review?** For information about this choice, including consent withdrawal, please see our Privacy Policy .

Reviewer #2: No

---

## [Author Response · Author response to Decision Letter 2]

17 Mar 2025

Dear Editors and Reviewers,

Thank you very much for carefully reviewing our manuscript and providing us with insightful, critical, and constructive feedback. Regarding the suggestions you have raised, we have a slight confusion and hope you can spare some valuable time to help us clarify.

In response to your request for the 'Response to Reviewers', 'Revised Manuscript with Track Changes', and 'Manuscript' files, we have uploaded these as separate files in our submission. However, we are unsure if we have uploaded them in the correct location or if there might be another reason that we have not fully understood your instructions. As a result, we are currently unclear on how to proceed with revising the manuscript at this stage.

We kindly ask for your patience and guidance in clarifying this matter. Could you please advise us on the next steps we should take to address the revisions? Please rest assured that we are fully committed to making the necessary changes with utmost care and rigor. Should you have any further questions or require additional information, please do not hesitate to contact us.

We look forward to your response and once again extend our sincere gratitude for your time and effort.

Best regards,

Feng-jie Xie

---

## [Editor Report · Decision Letter 2]

25 Mar 2025

The effect of lidocaine against sepsis-induced acute lung injury in a mouse model through the JAK2/STAT3 pathway

PONE-D-24-28154R2

Dear Dr. Feng-Jie,

We’re pleased to inform you that your manuscript has been judged scientifically suitable for publication and will be formally accepted for publication once it meets all outstanding technical requirements.

Kind regards,

Yong Jiang, Ph.D.

Academic Editor

PLOS ONE
---

## [Editor Report · Acceptance letter]

PONE-D-24-28154R2

PLOS ONE

Dear Dr. Xie,

I'm pleased to inform you that your manuscript has been deemed suitable for publication in PLOS ONE. Congratulations! Your manuscript is now being handed over to our production team.

Kind regards,

on behalf of

Professor Yong Jiang

Academic Editor

PLOS ONE